# Evaluation of Hematological, Biochemical Profiles and Molecular Detection of Envelope Gene (gp-41) in Human Immunodeficiency Virus (HIV) among Newly Diagnosed Patients

**DOI:** 10.3390/medicina59010093

**Published:** 2022-12-31

**Authors:** Asfa Anjum, Abaid ur Rehman, Hina Siddique, Ali A. Rabaan, Saad Alhumaid, Mohammed Garout, Souad A. Almuthree, Muhammad A. Halwani, Safaa A. Turkistani, Haitham Qutob, Hawra Albayat, Mohammed Aljeldah, Basim R. Al Shammari, Fatimah S. Alshahrani, Ali S. Alghamdi, Sami M. Alduwaihi, Adil A. Alibraheem, Shah Zeb, Basit Zeshan

**Affiliations:** 1Department of Medical Education, University of Lahore, Lahore 54590, Pakistan; 2Department of Medical Education, Sheikh Zayed Medical College, Rahim Yar Khan 06426, Pakistan; 3Department of Medical Education, Fatima Jinnah Medical University, Lahore 54000, Pakistan; 4Department of Public Health and Nutrition, The University of Haripur, Haripur 22610, Pakistan; 5Molecular Diagnostic Laboratory, Johns Hopkins Aramco Healthcare, Dhahran 31311, Saudi Arabia; 6College of Medicine, Alfaisal University, Riyadh 11533, Saudi Arabia; 7Administration of Pharmaceutical Care, Al-Ahsa Health Cluster, Ministry of Health, Al-Ahsa 31982, Saudi Arabia; 8Department of Community Medicine and Health Care for Pilgrims, Faculty of Medicine, Umm Al-Qura University, Makkah 21955, Saudi Arabia; 9Department of Infectious Disease, King Abdullah Medical City, Makkah 43442, Saudi Arabia; 10Department of Medical Microbiology, Faculty of Medicine, Al Baha University, Al Baha 4781, Saudi Arabia; 11Fakeeh College for Medical Science, Jeddah 21134, Saudi Arabia; 12Department of Medical Laboratory Technology, Faculty of Applied Medical Sciences, King Abdulaziz University, Rabigh 25732, Saudi Arabia; 13Infectious Disease Department, King Saud Medical City, Riyadh 7790, Saudi Arabia; 14Department of Clinical Laboratory Sciences, College of Applied Medical Sciences, University of Hafr Al Batin, Hafr Al Batin 39831, Saudi Arabia; 15Department of Internal Medicine, College of Medicine, King Saud University, Riyadh 11362, Saudi Arabia; 16Division of Infectious Diseases, Department of Internal Medicine, College of Medicine, King Saud University and King Saud University Medical City, Riyadh 11451, Saudi Arabia; 17Diagnostic Laboratory, Prince Sultan Military Medical City, Riyadh 12477, Saudi Arabia; 18ENT Department, Prince Sultan Military Medical City, Riyadh 12477, Saudi Arabia; 19Department of Microbiology, Faculty of Biomedical and Health Science, The University of Haripur, Haripur 22610, Pakistan; 20Department of Microbiology, Faculty of Life Sciences, University of Central Punjab, Lahore 54000, Pakistan; 21Faculty of Sustainable Agriculture, University Malaysia Sabah, Sandakan Campus, Locked Bag No. 3, Sandakan 90509, Sabah, Malaysia

**Keywords:** AIDS, Pakistan, retrovirus, communicable diseases, blood-borne diseases, risk-factors, HCV-HIV co-infection, HBV-HIV co-infection

## Abstract

The Human Immunodeficiency Virus (HIV) is a highly morphic, retrovirus that rapidly evolves through mutation as well as recombination. Because of the immunocompromised status in HIV patients, there is often a higher chance of acquiring different secondary infections followed by liver cirrhosis, hepatitis B & C, and HIV-associated nephropathy. The current study was conducted to see the prevalence of secondary infections, hematological and biochemical markers for liver and renal associated diseases, and to detect the envelope gene (GP41) in newly diagnosed HIV patients. A total of 37 samples were collected from HIV-positive patients registered in different hospital settings under the National AIDS control program. The collected samples were processed for hepatitis B, hepatitis C, hematological analysis, and biochemical analysis. To identify the envelope gene in newly diagnosed HIV patients, polymerase chain reaction (PCR) was performed using four gene-specific primers. The HIV infections were seen more in male as compared to females. A significant decrease in complete blood count was observed in HIV patients when compared to healthy individuals. There was a significant increase in aspartate aminotransferase (AST), alanine aminotransferase (ALT), urea, and creatinine observed in HIV patients. No significant difference was observed in alkaline phosphatase (ALP), total bilirubin, and albumin levels when compared to healthy control. Anemia was observed in 59.4% of HIV patients. A total of three (8.1%) patients were found to be co-infected with hepatitis B and one (2.7 %) was co-infected with hepatitis C. Out of these 37 tested samples, a total of four showed the successful amplification of the envelope gene. This study provides platform for the health care facilitators to regularly monitor the signs, symptoms and clinical biomarkers of HIV-associated infections to prevent toxicity at an early stage to improve the quality of life (QoL) and minimize the mortality rate in HIV patients. Envelope gene mutating frequently results in drug resistance, and thus future research on polymorphism analysis will reveal points of substitutions to improve drug designing.

## 1. Introduction

The human immunodeficiency virus (HIV) belongs to the lentivirus family (Ret-rovirus) that develops acquired immune deficiency syndrome (AIDS), a condition where the body’s immune system gradually weakens. This immunocompromised stage may lead to the appearance of lethal opportunistic infections (OIs) and possibility of cancer development [1]. According to an estimate, HIV has infected over 70 million individuals and killed almost 35 million people since its emergence [2,3]. Such stigma and prejudice experienced by individuals being HIV-positive and those at a greater risk of acquiring HIV has fueled shortages in HIV treatment, screening, and drug treatment [4].

HIV infections are also linked to a variety of hematological problems, particularly bone marrow defects as well as peripheral cytopenia [5,6]. Thrombocytopenia, anemia, and leucopenia are linked to a faster onset of AIDS. Direct HIV infection of hematopoietic precursor cells, bone marrow endothelial cells, as well as the inhibitory action of cytokines released following HIV infection and dissemination are among the methods that limit hematopoietic cell lines [3].

HIV infection promotes an increased prevalence of chronic HBV infection, liver cirrhosis, and tuberculosis infections [7]. Apart from these infections the discrepancies in parameters such as serum urea, creatinine, albumin (ALB), ALT, bilirubin, AST, and alkaline phosphatase (ALP) are prevalent. The use of antiretroviral treatment (ART) can have a negative influence on these conditions as well as increase the risk [8,9].

The HIV-1 epidemic in Pakistan continues to be a significant health risk for the public [10,11,12,13,14]. Since the encounter of the first case of HIV-1 infection in 1987, a continuous increase in the number of infected people has been observed. Initially, it was limited to isolated instances, but recently it has turned to outbreaks and concentrated epidemics. By the middle of 2020, the total number of infected people in Pakistan was 180,000 [15].

End-stage liver disease (ESLD), liver cirrhosis, and hepatocellular carcinoma (HC) appear to be the same blood indicators in HIV-HBV co-infections as they are in HIV-HCV co-infections [16,17]. The HIV co-infection with HBV and HCV constitutes a global burden, increasing the risk of morbidity and death among individuals who are exposed. Early identification of HBV/HCV-HIV co-infection is required for prompt and efficient therapy [18].

Envelope gene is the structural gene of HIV composed of surface glycoprotein GP120 and transmembrane glycoprotein GP41. The viral transmembrane glycoprotein (GP41) is the most important immunogenic protein for HIV sero-detection. Antibodies against the transmembrane glycoprotein are among the first to emerge following an HIV infection’s seroconversion, and they stay quite robust throughout the asymptomatic and symptomatic stages. HIV fully depends on these glycoproteins to enter host cells (CD4+ T-cells) [19].

HIV/AIDS persists as a considerable global health problem [20]. Approximately 37 years after the first report of acquired immunodeficiency syndrome (AIDS), adequate control of the AIDS pandemic remains ambiguous. In terms of this concern, the molecular pathophysiology of HIV-1 is crucial, a virus that has developed a variety of ways to elude immune control [21,22]. Drugs have been designed which inhibit the fusion or entry of HIV to the host cells but, to date, no drug available can fully eliminate the HIV infection from the patient due to its high mutation rate and genetic diversity [23,24]. The current study was designed to identify the GP41 gene of HIV among newly diagnosed patients. Keeping in mind the worse health related issues in HIV-positive cases, the current study was designed to evaluate the hematological and biochemical parameters, to screen the samples for secondary co-infections, and to identify the GP41 gene of HIV among newly diagnosed patients.

## 2. Materials and Methods

### 2.1. Study Design and Ethical Consideration

The study was conducted in accordance with the Declaration of Helsinki and approved by the Institutional Review Board of the University of Haripur (UOH/DASR/2021/4955). Each of the study’s participants was told of the study’s aim and objectives first and then a written informed consent form was obtained from each patient who participated. The current study was conducted by the Department of Microbiology, The University of Haripur in collaboration with Department of Microbiology, University of Central Punjab, and the Department of Medical Education, University of Lahore from February 2021 to May 2021. The molecular analysis was carried out at the Department of Microbiology, University of Central Punjab.

### 2.2. Consent Development

A Pictorial Questionnaire (PQ) was used in the current study to obtain the demographical data of studied patients. The study subjects were also interviewed regarding their level of education, marital status, employment status, sexual behavior, appearance of common symptoms, and the possible history for acquiring the infection. All the data and samples were kept confidential and anonymous. Informed written consent was received from all study subjects participating in this study. The patients were administered free diagnostic tests such as complete blood count (CBC), liver function tests (LFTs), renal function tests (RFTs), and screening tests for hepatitis B and C.

### 2.3. Samples Collection

Blood samples from newly diagnosed HIV-positive patients (*n* = 37) were collected from different diagnostic centers linked with the National AIDS Control Program (NACP) after getting the signed informed consents from the patients. To compare the results of hematological and biochemical parameters with HIV patients, blood samples were collected from a total of 20 healthy controls. The healthy controls were selected based on their health status. Before recruiting the healthy controls for current study, these individuals were questioned about their medical history and sample collection proceeded only if they were found healthy and not suffering from any diseases.

#### 2.3.1. Inclusion Criteria for HIV Patients

Newly diagnosed HIV patients aged 18 years or older with CD4 cell counts more than of 200 cells/mm^3^, viral load of more than 1000 copies/mL, and accessibility to past and current medical and laboratory records were included.

#### 2.3.2. Exclusion Criteria for HIV Patients

Those age 18-years or younger were excluded as they are less prevalent for HIV positivity in Pakistan, CD4 cell counts less than 200 cells/mm^3^ that are vulnerable to get bacterial and fungal opportunistic infections, and HIV viral loads less than 1000 copies mL.

### 2.4. Hematological and Biochemical Analysis

A hematology analyzer (Sysmex KX-21) was used for the analysis of CBC, employing 2 mL of whole blood (hemoglobin, total leukocytes count, hematocrit, and platelet count). For the analysis of the complete blood count, EDTA-containing vials were used. Before the process, all the samples were placed on the roller for 5 min at room temperature. As the samples mixed, CBC vials were introduced to the chamber of the Sysmex KX-21 for analysis of the complete blood counts.

The normal reference range of hemoglobin was 13–18 g/dL in males and 12–15 g/dL in females. The total leukocyte count normal reference range was 2.6–8.3 × 10³ mm³. Some control patients had borderline leukocyte counts due to infections other than HIV infection, i.e., seasonal fever. The normal reference range for platelet count was 140–440 × 10³ mm³. The Biochemical tests like LFTs and RFTs were performed on Biosystems BA200, a fully automated clinical chemistry analyzer.

### 2.5. Serological Screening

#### 2.5.1. Alere Determine Rapid Device

The initial screening of HIV was performed on blood samples using Alere Determine™ HIV 1/2 Ag/Ab Combo rapid devices. These rapid devices can identify antibodies against HIV-1 & 2 and the HIV-1 antigen.

#### 2.5.2. Chemiluminescent Microparticle Immuno-Assay (CMIA) for HIV-1/2

For the confirmation of screening performed through the Alere rapid device, a chemiluminescent microparticle immuno-assay (CMIA) was employed. The CMIA was performed on Abbott’s Architect i1000 SR HIV Ag/Ab Combo assay which is used for the concurrent quantitative analysis of HIV p24 antigen and antibodies to HIV in human blood. All reactive specimens were centrifuged (10,000 rpm for 10 min) and retested.

#### 2.5.3. CMIA for HBsAg

The CMIA was performed on Abbott’s Architect i1000 SR assay which was used for the concurrent quantitative analysis of surface antigen (HBsAg) for the hepatitis B virus in human blood. All reactive specimens were centrifuged (10,000 rpm for 10 min) and retested.

#### 2.5.4. CMIA for Anti-HCV

The CMIA was performed on Abbott’s Architect i1000 SR assay which was used for the concurrent quantitative analysis of antibodies to HCV in human blood. All reactive specimens were centrifuged (10,000 rpm for 10 min) and retested.

### 2.6. Molecular Identification

#### 2.6.1. RNA Extraction

QIAamp Viral RNA Mini Kit (Qiagen, Valencia, CA, USA) catalog No. 740956 was used to extract viral RNA from HIV-positive individuals’ blood samples. Out of 37, RNA was extracted from 16 samples and quality of RNA was assessed by 1% agarose gel electrophoresis. Extracted RNA was quantified by using NanoDrop (Optizen, Mecasys, Daejeon, Korea) and readings were recorded at wavelengths 260/280.

#### 2.6.2. cDNA Synthesis

The HIVCR1 reverse primer was used for the synthesis of cDNA. The primer was “TGCTAGAGATTTTCCACACTGAC” with length of 23 bp and 44% GC content. The melting temperature was 61 °C.

Following RNA extraction, complete viral RNA, reaction buffer, an enzyme reverse transcriptase, a primer, dNTPs, and RNase inhibitor were used to reverse-transcribe the RNA into complementary DNA (cDNA) using a commercially available kit (Solis BioDyne’s FIREScript^®^ RT cDNA Synthesis kit). The resulting cDNA was then used in a PCR reaction.

To prepare cDNA, the FIREScript^®^ RT cDNA synthesis kit was used. The cDNA synthesis procedure was as follows:First, thawed, mixed, and briefly centrifuged the components of the kit and stored them on ice.Then RNA template (10 µL), primer (1 µL), and nuclease-free water (1 µL) was added into a sterile, nuclease-free tube on ice.The template RNA (2 µL), reverse primer (1 µL), and nuclease free water (13 µL) were added in a screw cap Eppendorf and were gently mixed, briefly centrifuged, and kept at 65 °C for 5 min. Cooled down over ice, spun down, and then re-chilled the vial.Then, 10× Reaction Buffer (2 µL), RiboLock RNase Inhibitor (0.5 µL), 10 mM dNTP Mix (1 µL), and FIREScript RT (0.5 µL) were added, gently mixed and centrifuged.Incubated for 5 min at 25 °C, 30 min at 60 °C.Terminated the reaction by heating at 85 °C for 5 min.The whole process was carried out in a Multigene Optimax thermal cycler (Labnet, Iselin, NJ, USA). After performing the cDNA synthesis protocol cDNA was used immediately for amplification of the desired gene or stored at −20 °C.

#### 2.6.3. Polymerase Chain Reaction

For the molecular characterization of HIV, a polymerase chain reaction was performed. Two pairs of primers with selected conditions were used for the identification of envelope gene of HIV.

In this study, the presence of the HIV envelope gene (GP41) was determined by using PCR. Four primers were used for the detection of glycoprotein GP41 as shown in Table 1. The sequences of the primers were designed by using Oligo Calculator and confirmed through Primer BLAST (http://www.hiv.lanl.gov/, accessed on 3 February 2021).

#### 2.6.4. Conditions for Polymerase Chain Reaction

Multigene Optimax thermal cycler (Labnet, Iselin, NJ, USA) was used to perform the PCR to detect GP41 of envelope gene glycoproteins of HIV-1. The final solution reaction mix of 25 μL was prepared for single PCR. 2 μL of cDNA along with the 5 μL of ready-to-use master mix (Solis BioDyne, Tartu, Estonia) was added. 1.5 μL of each forward and reverse primer were also added. In the last 15 μL of the autoclaved PCR water was added to prepare the final volume of 25 μL. Short spin for through mixing was given in centrifuge at 5000 rpm for 30 s. The following conditions were applied to perform PCR amplification: Initial denaturation at 95 °C for 5 min, 40 cycles each of 95 °C for 45 s, followed by varying annealing temperatures for 30–45 s, primer initial extension at 72 °C at 1 min, and final extension at 72 °C for 10 min.

#### 2.6.5. Gel Electrophoresis

Agarose Gel Electrophoresis separates fragments of DNA based on size and charge by using the MultiSub Mini Horizontal Electrophoresis System. Two grams of agarose powder were dissolved in 100 mL of 1× tris boric acid EDTA (TBE) buffer to make agarose gel. After that, the beaker was heated on a hot plate for 1 min. For the visualization of DNA bands, 5 µL of ethidium bromide was mixed once it had cooled. Finally, the gel was poured into the caster containing combs and left for 20 min to settle. After solidification, combs were removed, and the caster was settled in the gel tank. DNA 5 µL was loaded into a well by mixing with 5 μL of loading dye. The electric current was applied for 50 min at 80 V and 300 Amp of current. After running, the gel was then observed with a gel documentation system (MicroDoc, Cleaver Scientific, Rugby, UK). Fragments of GP41 were seen at 494 bp and 552 bp using a 100 bp DNA ladder.

### 2.7. Statistical Analysis

The data was entered in SPSS version 26.0 (IBM, New York, NY, USA). The descriptive analysis was applied to check the frequency (n), percentage (%), mean, and standard deviation (SD). The chi-square test was run to see the difference among the HIV-positive patients and healthy controls. A *p*-value of <0.05 was considered statistically significant.

## 3. Results

### 3.1. Study Design

Thirty-seven (37) blood samples of HIV-positive patients were collected from different institutes linked with National AIDS Control Program. Twenty (20) normal patients were also included as controls.

### 3.2. Demographic Charateristics of Studied Patients

The study participants were comprised of 41 males, 16 females, and no transgender individuals participated due to interpersonal and structural barriers to health services. In HIV-infected patients, the proportion of males (*n* = 24, 64.86%) in this study was higher than females (*n* = 13, 35.13%). Majority of the positive cases were found in the patients with age of >45 years (43.24%).

In 37 HIV-infected patients, four were found to be infected with viral hepatitis, out of which three were detected as HBV infected and one was co-infected with HCV. The demographic characteristics of study participants is given in Table 2.

There were three patients seen who had history of sexual intercourse with both men (men with men/gay) and transgenders. The history of transmission and the stage of HIV infection of the studied patients is given in Table 3.

The prevalence of common symptoms appeared in studied HIV patients is given in Table 4. Few patients had shown more types of symptoms together. The most common symptom observed in studied patients was the weakness, weight loss, and body pain.

The mean ± SD hemoglobin level in HIV patients was 10.01 ± 0.77 mg/dL and the mean ± SD total leucocyte count was 7.36 ± 1.01 × 10³ mm. While the mean ± SD total platelet count was 183.51 ± 38.13 × 10³ mm.

### 3.3. Comparison of CBC of HIV-Positive Patients and Healthy Controls

Complete blood count includes hemoglobin (Hb), total leukocyte count (TLC), and platelet count (PLT). Cytopenia (decreased Hb, TLC, and PLT count than normal range) is a common hematological abnormality in HIV and is a predictor of illness formation and fatality. The findings of this study suggest that there is a general decrease in the prevalence of anemia (Hb), leukopenia (TLC), and thrombocytopenia (PLT) (Table 5).

### 3.4. Prevalence of LFTs and RFTs in HIV Patients and Healthy Controls

These liver enzymes were assessed to evaluate the HIV-induced hepatic injury. The level of ALT and AST was significantly increased in HIV patients as compared to control patients. No significant difference was found in HIV-positive patients and the control population as shown in Table 6.

### 3.5. Prevalence of Anemia in HIV Patients and Healthy Controls

The presence of anemia (Hb < 10.0 g/dL) is seen to be greater in HIV-positive patients. HIV infection alone, without any chronic condition, can cause anemia. The level of blood immunoreactive erythropoietin in HIV patients did not rise in lockstep with progressive anemia. In the current study, about 59.4 % HIV patients were found to be anemic as shown in Table 7.

### 3.6. Prevalence of HBV and HCV Co-infection in HIV Patients and Controls

Despite significant advances, and preventative and therapeutic approaches, co-infection of HIV-positive persons with HBV and HCV is an increasing worldwide concern. The HIV, HBV, and HCV are all blood-borne viruses with almost similar transmission modes. The effect of ART on survival in high-income countries has revealed chronic liver disease from HBV or HCV as a leading cause of morbidity and death in people living with HIV. The overall viral hepatitis co-infection was 10.8 %. Three (8.1 %) detected positive for HBV while one (2.7 %) was positive to HCV and none of the HIV patients recorded as triple infection in 37 HIV-positive patients (Table 8).

### 3.7. Polymerase Chain Reaction of GP41 Fragment 1 and 2

The concentration of RNA isolated from all HIV-positive subjects is shown in Table 5. The total number of samples proceeded for RNA extraction and to be analyzed for detection of envelope gene (GP41) was 37.

From the PCR analysis of 37 HIV-positive patients, the four patients were identified with envelope gene (GP41) in their genome. The HIV-positive patient specimens were observed for HIV envelope (GP41) in which 80.2% were seen negative for HIV envelope (GP41) gene primers, whereas (10.8%) of HIV-positive specimens were seen positive for envelope gene GP41. The gel electrophoresis of RNA, GP41 Fragment 1 and GP41 Fragment 2 is shown in Figure 1.

## 4. Discussion

The first and crucial step in the prevention strategies of HIV infection is the diagnosis [25,26]. Antibodies generated in reaction to HIV developed in a span of 1-to-4 months after infection [27]. Analysis for the detection of HIV begins with an enzyme-linked immunosorbent assay (ELISA), a test for antibody detection. The goals for HIV antibody testing include safety during transfusion, donation or transplant procedures, follow-up of HIV infection, and final diagnosis [28,29]. In addition to this, the Western blot technique is also performed as confirmatory testing, and PCR is preferred to monitor antiretroviral therapy and HIV detection in newborns [15,30]. The current study was designed to evaluate the hematological and biochemical parameters, to screen the samples for secondary co-infections, and to analyze the presence of GP41 among newly diagnosed HIV patients.

Last year an outbreak of HIV infection was observed in Larkana, Pakistan that led to a concentrated epidemic countrywide showing a 57% increase in the overall infected population. Biomarker abnormalities were widely prevalent in people living with HIV showing a range from 20 to 93% that could lead to renal dysfunction and liver dysfunction, accounting for almost 14–18% of all deaths [11].

A study showed the proportion of males (73.5%) was higher than females (20.9%) and transgenders (5.6%) [10]. An earlier study in Pakistan found a similar age range of 30–39 years as dominating, with a high rate of HIV infection (30%) [11]. Hematopoietic progenitor cells suppression by HIV was a leading cause of decreased Hb in HIV patients [31]. A study established that HIV could worsen the fate by accelerating the onset of AIDS, causing liver damage, and decreasing immune recovery, reported mean urea, and creatinine levels were relatively higher in HIV-negative patients than in HIV-positive individuals [32].

A previous study established that HIV could worsen the fate by accelerating the onset of AIDS, causing liver damage, and decreasing immune recovery. The current results of this analysis are significant because the study population had a high frequency of mild renal impairment. Many studies have revealed that HIV patients are more susceptible to developing renal illness. The mean urea and creatinine levels were relatively higher in HIV-negative patients than in HIV-positive individuals [21]. Malnutrition is one of the major causes of anemia in HIV-infected patients [33]. This finding is similar to our 2010 study showing a 65% prevalence of anemia. They showed decreased red cell production as a cause of anemia [29].

An overall rate of 8.5%, comprising 7% HBV and 1.5% was observed. They have greater levels of ALT, AST, and ALP, which makes them susceptible to hepatic injury. They demonstrated a poor immunological response, poor therapeutic response, and severe liver toxicity as a factor for HBV and HCV co-infection [34]. This significant HBV-infection incidence confirms previous findings in Ghana’s Eastern (8.8%) and Central regions (6.1%) in addition to many other African countries; however, the figures found in other sections of the nation were slightly lower [5]. The low figure of HCV-confirmed samples in our research might be attributed to low donation rates and the absence of drug abusers across study subjects. Both are strong predictors for HCV transmission between HIV patients. Hence, we were unable to detect any of the people who had triple infections. This is also in line with previous research in Nigeria, Ghana [35].

The compensatory mutations in the envelope gene render the amino acid substitutions conferring broad escape from defects in virus replication. It also mediates highly efficient cell-to-cell transmission [36]. It is also reported in a study that defective mutants are produced by recombination in the envelope region. This promotes the phylogenetic evolution of the viruses increase in diversity of the virus population. The role of defective genes may be converted from junk to useful ones [19,36].

Several drugs have been designed previously to control the progression of the disease caused by infection through HIV-1 [31,37]. The envelope gene has been a novel target of the first class of antiretroviral drugs against HIV-1 and has been studied extensively worldwide [23,38]. HIV infection is also linked to a variety of hematological problems, particularly bone marrow defects as well as peripheral cytopenia [37]. Thrombocytopenia, anemia, and leucopenia are linked to a faster onset of AIDS and an even worse chance of survival [38].

In the current study, 37 samples were identified as HIV-positive by CMIA method and confirmed by PCR. Out of these 37 isolates, four (10.8%) carried the GP41 gene. GP41 of HIV1 was divided into two fragments of 494 bp and 552 bp for PCR amplification. The molecular identification of GP41 of HIV-1 allowed us to partially understand the genetic makeup of the surface gene. New therapies may provide hope for increased longevity among people living with AIDS, but the biology of HIV virus presents significant obstacles to finding a new cure and/or vaccine. The development of new treatment options may provide hope of longer life to HIV-infected patients, but the existence of rapid mutation is still a challenge in developing a vaccine for a better cure.

**Study limitations:** One of the major limitations of the current study was its time duration. Because of pandemic COVID-19 it was very difficult to conduct a study for a longer period. Because of the small sample size, the current study was unable to extrapolate the findings to the whole community/population. Another limitation was that there might be some missing data as the questionnaire was distributed to the patients for collection of their health-related data and some of the patients would not agree to discuss their medical conditions with an interviewer. Hence, when interpreting the results, these limitations should be taken into an account.

## 5. Conclusions

The results of the current study could be helpful for the pharmaceutical companies and micro and molecular biologists to conduct further large-scale studies on subtyping, sequencing, distribution, and drug designing. The envelope gene (GP41) contains regions that mutate frequently and cause drug resistance; therefore, in future studies the mutational analysis of the gene and its molecular information could reveal the points of mutation and further improve drug designing. Furthermore, the data of clinical biomarkers from this study could help health care facilitators to keep track of variations in blood parameters.

## Figures and Tables

**Figure 1 medicina-59-00093-f001:**
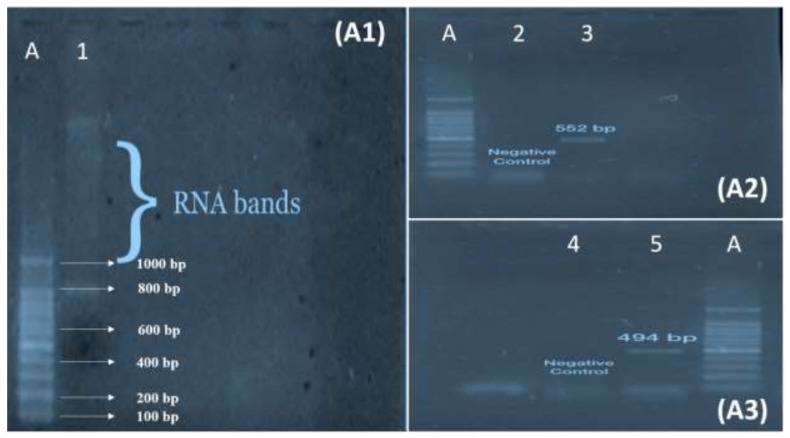
A: 100 bp RNA Ladder. (**A1**): Agarose 1% gel electrophoresis of HIV-1 RNA. 1: Bands of RNA. (**A2**): Agarose 2 % gel electrophoresis of glycoprotein GP41 F2 of HIV Envelope gene PCR products. 2: Negative Control. 3: Positive bands for 2nd fragment of GP41. (**A3**): Agarose 2 % gel electrophoresis of glycoprotein GP41 F1 of HIV Envelope gene PCR products. 4: Negative Control. 5: Positive bands for 1st fragment of GP41.

**Table 1 medicina-59-00093-t001:** PCR primers for the identification of envelope gene (GP41) of HIV.

Name	Primer Sequence	Product Size (bp)	Annealing Temperature (°C)
TMF1	CATGGGCCAAGTTCCGAGC	494	59
TMR1	CGGACATCGGGAGGAGC		
TMF2	GGGGGGTACTAGGAACACG	552	56
TMR2	CCCTCGAAGGTGGATCGAG		

**Table 2 medicina-59-00093-t002:** The demographic characteristics of studied HIV patients (*n* = 37).

Characteristics		Frequency (*n*)	Percentage (%)
Gender	Male	24	64.86
	Female	13	35.13
Age (Years)	18–34	8	21.62
	35–44	13	35.13
	>45	16	43.24
Marital status	Married	33	89.18
	Unmarried	4	10.81
Employment status	Employed	20	54.05
	Unemployed	3	8.10
	Self-business	14	37.83
Education level	Primary school certificate	17	45.94
	Secondary school certificate	16	43.24
	Bachelor’s degree	4	10.81

**Table 3 medicina-59-00093-t003:** The clinical history of studied HIV patients (*n* = 37).

Characteristics		Frequency (*n*)	Percentage (%)
Possible route of acquiring HIV infection	Sexual intercourse with men	05	13.51
	Sexual intercourse with women (sex workers)	16	43.24
	Sexual intercourse with transgender	06	16.21
	Intravenous drug use	06	16.21
	Reusing of injection needle from infected person	01	2.70
	Parental transfer	01	2.70
	Sexual intercourse with spouse	03	8.10
	Unknown	02	5.40
Stage of HIV infection	Symptomatic	21	56.75
	Non-symptomatic	16	43.24
	Co-infected with tuberculosis	04	10.81

**Table 4 medicina-59-00093-t004:** The prevalence of common symptoms in studied HIV patients (*n* = 37).

Symptoms		Frequency (*n*)	Percentage (%)
Psychopathological symptoms	Anxiety	11	29.72
	Depression	05	13.51
Physical symptoms	Weakness	32	86.48
	Weight loss	29	78.37
	Lack of energy	16	43.24
	Lack of hunger	19	51.35
	Diarrhea	04	10.81
	Body pain	27	72.97
	Hair loss	05	13.51
	Dizziness	06	16.21
	Respiratory problems	06	16.21
	Difficulty in walking	24	64.86
	Tiredness	30	81.08
	Discharge from genitalia	04	10.81

**Table 5 medicina-59-00093-t005:** Comparison of hematological parameters between HIV patients and healthy controls.

Parameter	Unit	Positive Patients	Normal Patients	*p*-Value
HB	g/dL	10.01 ± 0.77	12.51 ± 0.60	<0.0001
TLC	10^3^ mm^3^	7.36 ± 1.01	8.22 ± 0.78	0.001
PLT	10^3^ mm^3^	183.51 ± 38.13	301.05 ± 39.34	<0.0001

**Table 6 medicina-59-00093-t006:** Comparison of biochemical parameters of liver function tests and renal function tests between HIV-positive patients and healthy controls.

Parameter	Unit	Positive Patients	Normal Patients	*p*-Value
ALT	U/L	45.51 ± 38.98	28.3 ± 3.31	0.0548
AST	U/L	32.35 ± 15.37714	28.4 ± 4.29	0.2669
Total Bilirubin (T. Bil)	mg/dL	0.8 ± 0.42961	0.71 ± 0.10	0.2258
ALP	U/L	97.05 ± 39.33	74.85 ± 8.29	0.0160
Albumin (Alb)	g/dL	4.3 ± 0.15	4.0 ± 0.14	<0.0001
Urea	mg/dL	38.4 ± 7.13	30.8 ± 3.08	<0.0001
Creatinine	mg/dL	1.0 ± 0.34	0.9 ± 0.14	0.3700

**Table 7 medicina-59-00093-t007:** Prevalence of anaemia in HIV-positive patients.

Total No of HIV-Positive Patients	HIV-Positive with HB < 10.0 g/dL	HIV-Positive with Hb > 10.0 g/dL
37	22 (59.4%)	15 (40.6%)

**Table 8 medicina-59-00093-t008:** Prevalence of HBV and HCV co-infections in HIV-positive patients.

Viral Hepatitis Co-Infection	Number (%)
Total HIV patients	37 (100 %)
Overall co-infection	4 (10.8 %)
HBV	3 (8.1 %)
HCV	1 (2.7 %)

## Data Availability

All of the data reported in this study has been reported in the current manuscript.

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
