# Peer review of "Evaluation of Hematological, Biochemical Profiles and Molecular Detection of Envelope Gene (gp-41) in Human Immunodeficiency Virus (HIV) among Newly Diagnosed Patients"

_medicina, 2022, doi:10.3390/medicina59010093_

Round 1

Reviewer 1 Report

Unfortunately, I am sorry to write a negative review. In my opinion, the article is not suitable for publication. This type of work would be interesting at the very beginning of the HIV epidemic, in the early 1980s. If the authors are looking for cheap markers for people who should be offered HIV testing in their country's population, they should also take into account other variables, such as selected information obtained during the anamnesis and make a model, and veryfie it for example with ROC. It would be also interesting a paper describing the population of infected people living in Pakistan - in which groups the infection occurs most often, what are the most common routes of transmission of the virus, whether the populations of HIV infected people in Pakistan differ from the population and the most common routes of infection in other countries and regions, etc.
Unfortunately, on the basis of the presented results, nothing can be said about the infected population in Pakistan.
Of the many objections I have, I will mention just a few:
1. extremely low number of patients and control group, which is associated with a high probability of a type II error. Such small numers of patients in study group is so surprising, since the study is multicentre, and according to the authors, there are nearly 180,000 infected people in Pakistan. In addition, it is surprising that there is such a large number of authors from three countries in the paper, and study has only 57 participants.
2. the statistical methods used are not described in the method section
3. there is no need to describe in detail the methods of determining commonly used biochemical parameters, such as aminotransferases or creatinine, etc. However, the selecting patients for the study was not described clearly - where they were from, when they were recruited for the study, how the control group was selected, whether they were treated with antiretroviral therapy e.t.c.
4. describing the study and control groups should be provided in the text and not as additional material, on the other hand figures 1 and 2 are unnecessary - they do not provide additional information
4. the discussion is inconsistent and sometimes incomprehensible (eg in the part devoted to the results it was mentioned that there were no transgender people in the study group, and the discussion mentioned 5.6% of them. Where? - in the study? In Pakistan? It is not clear.). The mention of SIV in the context of the work itself is also incomprehensible. Discussion of the results adds nothing. Why only two african countries were choose? Are they similar to Pakistan?
5. the part devoted to discussing the limitations of the study is missing

Author Response

Reviewer 1

Comments and Suggestions for Authors

Unfortunately, I am sorry to write a negative review. In my opinion, the article is not suitable for publication. This type of work would be interesting at the very beginning of the HIV epidemic, in the early 1980s. If the authors are looking for cheap markers for people who should be offered HIV testing in their country's population, they should also take into account other variables, such as selected information obtained during the anamnesis and make a model, and verify it for example with ROC.

Response: Dear reviewer, we would like to appreciate that you have provided us with your kind comments and suggestions. We agree that the last version of manuscript was not suitable for publication because of the lack of important data. That’s why we would like to appreciate that, after addressing comments from you and other reviewers, the manuscript has been significantly improved. Furthermore, we have added some molecular data in the revised version of manuscript.

It would be also interesting a paper describing the population of infected people living in Pakistan - in which groups the infection occurs most often, what are the most common routes of transmission of the virus, whether the populations of HIV infected people in Pakistan differ from the population and the most common routes of infection in other countries and regions, etc. Unfortunately, on the basis of the presented results, nothing can be said about the infected population in Pakistan.

Response: Line 277-290, table 2-4: Dear reviewer, thank you for your valuable suggestion. The respective new information has been added in the revised version of manuscript.

Of the many objections I have, I will mention just a few:

  1. extremely low number of patients and control group, which is associated with a high probability of a type II error. Such small numers of patients in study group is so surprising, since the study is multicentre, and according to the authors, there are nearly 180,000 infected people in Pakistan. In addition, it is surprising that there is such a large number of authors from three countries in the paper, and study has only 57 participants.

Response: Dear reviewer, we agree that as compare to the infected population, the sample size in the current study is extremely low. The inclusion criteria for the current study were to include the newly diagnosed patients only. The patients who were already doing their regular check-ups were not included in the current study. We hope that you will understand the inclusion criteria and will allow us to proceed further.

  1. The statistical methods used are not described in the method section.

Response: Section 2.7, Line 254-257: The statistical analysis has been added in the revised version of manuscript.

  1. there is no need to describe in detail the methods of determining commonly used biochemical parameters, such as aminotransferases or creatinine, etc. However, the selecting patients for the study was not described clearly - where they were from, when they were recruited for the study, how the control group was selected, whether they were treated with antiretroviral therapy e.t.c.

Response: The extra-detailed information about the tests has been removed. Furthermore, the material and methods section has been revised. The description about healthy controls has been added in the revised version of manuscript (Line 161-165). The inclusion and exclusion criteria for HIV patients has been specified also.

  1. describing the study and control groups should be provided in the text and not as additional material, on the other hand figures 1 and 2 are unnecessary - they do not provide additional information.

Response: Line 139-144: The description about healthy controls has been added in the revised version of manuscript. Furthermore, the figure 1 and 2 from the last version of manuscript have been removed from the revised version of manuscript.

  1. the discussion is inconsistent and sometimes incomprehensible (eg in the part devoted to the results it was mentioned that there were no transgender people in the study group, and the discussion mentioned 5.6% of them. Where? - in the study? In Pakistan? It is not clear.). The mention of SIV in the context of the work itself is also incomprehensible. Discussion of the results adds nothing. Why only two african countries were choose? Are they similar to Pakistan?

Response: Line 339-346, 381-410: The discussion section has been revised and more reports have been discussed in the revised version of manuscript.

  1. the part devoted to discussing the limitations of the study is missing

Response: Line 403-410: The study limitations has been added in the revised version of manuscript.

Reviewer 2 Report

Dear Sir,

I have gone through the manuscript entitled, Evaluation of Blood indicators for HIV-associated Pancytopenia, Nephropathy, and Liver disorders followed by Hepatitis B & C co-infections” carefully.

It is a fair attempt but not presented clearly. The manuscript needs some revisions. There are some concerns which need to be addressed. The comments are given below :

Comments :

1.        There are few errors that have been highlighted in yellow pop-up notes in the manuscript.

2.        Some sentences / words requiring changes have been highlighted.

3.        The authors should keep in mind that the paper will read by many across the world.

4.        Throughout the manuscript, there are several grammatical errors.   

5.        Authors : There are 19 authors and 20 Institutes. The study and analyses have been carried out in Haripur, Pakistan. I fail to understand what is the contribution of other Institutes [Saudi Arabia -15, Jeddah - 1, Malaysia - 1] mentioned in the first page. Have the authors really contributed in the study or just for the sake of friendship, the corresponding author, Dr. Basit Zeshan has given their names. For only 37 HIV positive samples, there are 19 authors. Very strange.

6.        The authors have reported the results of blood tests of only 37 HIV positive samples and very few markers like CBC, LFTs, RFTs.

7.        Abstract : Keywords : Give precise relevant words, at least six. Include the place of work, i.e., Pakistan, it is important.

8.        Introduction : is too short.

9.        Declaration : The authors state that Ethics committee of Haripur has approved the study.

10.    Methodology : Duration is February to May,2021. Is it only 4 months ?

11.    Results : not described properly. There is no need of controls at all. Just give the co-infection details and other parameters in the HIV positive patients. Several other socio-demographic characteristics, risk factors and clinical profile could have been added.

12.    Table : needs to be modified. There are 4 tables that too have not been given the right headings. The same information has been repeated in the graphs. In fact, there is no need of coloured graphs if the information is given in tables. Such colourful graphs are for power point presentations.

13.    Discussion : is short without any referral to other reports.

14.    Conclusion : in this section, the authors state that clinical indicators for HIV-associated secondary infections were analyzed in 57 individuals (37 HIV-positive & 20 controls). Do they expect the infections in normal controls ?

15.    References : There are many irrelevant references and there is no uniformity. Give at least 35-40. Give the names of authors, volume and page numbers, of cited references.

16.    Acknowledgements : As this is a patient centred research article, there would be a team of Lab. Technicians and other staff doing the several investigations and other follow up tests. Please include their names in the acknowledgement section.

17.    I would suggest the authors to read the Instructions to Authors carefully before submitting any article to a journal.

Keeping in view the impact factor of the journal, wide readership, I feel the manuscript needs major revisions at this stage.

If the authors can incorporate the suggested changes in the article, it would be better.

These suggestions are meant to improve the manuscript in a good journal.

Author Response

Reviewer 2

Comments and Suggestions for Authors

Dear Sir,

I have gone through the manuscript entitled“Evaluation of Blood indicators for HIV-associated Pancytopenia, Nephropathy, and Liver disorders followed by Hepatitis B & C co-infections” carefully. It is a fair attempt but not presented clearly. The manuscript needs some revisions. There are some concerns which need to be addressed. The comments are given below:

Response: Dear reviewer, we would like to appreciate that you have provided us with your kind comments and suggestions. We agree that the last version of manuscript was lacking of important data. That’s why we would like to appreciate that, after addressing comments from you and other reviewers, the manuscript has been significantly improved. Furthermore, we have added some molecular data in the revised version of manuscript.

Comments:

  1. There are few errors that have been highlighted in yellow pop-up notes in the manuscript.

Response: The yellow highlighted parts in the provided PDF file has been addressed and corrected in the revised version of manuscript.

  1. Some sentences / words requiring changes have been highlighted.

Response: Dear reviewer, the changes has been made in the revised version of manuscript and highlighted in red font colour.

  1. The authors should keep in mind that the paper will read by many across the world.

Response: Dear reviewer, we would like to really appreciate that you have provided us with your very valuable suggestions. The manuscript has been revised according to the comments from you and other reviewers. And we hope that, the changes will attract the readership and will be suitable internationally.

  1. Throughout the manuscript, there are several grammatical errors. 

Response: The manuscript has been thoroughly revised for English proofreading and grammatical mistakes.

  1. Authors:There are 19 authors and 20 Institutes. The study and analyses have been carried out in Haripur, Pakistan. I fail to understand what is the contribution of other Institutes [Saudi Arabia -15, Jeddah - 1, Malaysia - 1] mentioned in the first page. Have the authors really contributed in the study or just for the sake of friendship, the corresponding author, Dr. Basit Zeshan has given their names. For only 37 HIV positive samples, there are 19 authors. Very strange.

Response: Dear reviewer, we would like to appreciate once again your kind comment on the authorship criteria. At first, we would like to mention about Dr. Basit Zeshan, the corresponding author, actually at the time of study, Dr. Basit was a faculty member in a University in Punjab, while recently he has been moved to UMS, Malaysia. The query about other authors, we would like to clarify that there is no favourable authorship at all in the study. All of the authors have participated in the study, individual funding for the expenses on current study and the funding for APC. Furthermore, the authorship contributions have been added in the revised version of manuscript. We agree that for this small sample size study, the 19 authors are more then enough, that’s why after reading comments from you, the 3 of authors agree to remove their names and put their names in the acknowledgment section. We hope that you will understand the authorship criteria, specifically in terms of individual funding, as in Pakistan it is very difficult for us to get the funding from government bodies because of severe financial issues in the country.

  1. The authors have reported the results of blood tests of only 37 HIV positive samples and very few markers like CBC, LFTs, RFTs.

Response: Dear reviewer, we agree that the reported results were less for the study in good impact factor journal like “Medicina”. We have added some more molecular based data in the revised version of manuscript. And hope that it will improve the overall significance of manuscript.

  1. Abstract: Keywords:Give precise relevant words, at least six. Include the place of work, i.e., Pakistan, it is important.

Response: More suitable keywords have been provided in the revised version of manuscript.

  1. Introduction: is too short.

Response: Line 67-74, 97-103, 108-115: More relevant background has been added in the revised version of manuscript.

  1. Declaration:The authors state that Ethics committee of Haripur has approved the study.

Response: Line 118-120: The statement has been revised.

  1. Methodology:Duration is February to May,2021. Is it only 4 months ?

Response: Dear reviewer, as we discussed earlier that our inclusion criteria was to include only newly diagnosed patients from the study area. We agree that if we increase the study duration, it might increase the sample size also. But as we know that because of pandemic COVID-19 it was very difficult to conduct a study for a longer period. It was one of the limitations of study and has been elaborated at the end of discussion section of the revised version of manuscript.

  1. Results: not described properly. There is no need of controls at all. Just give the co-infection details and other parameters in the HIV positive patients.Several other socio-demographic characteristics, risk factors and clinical profile could have been added.

Response: Dear reviewer, unfortunately we are not able to remove the healthy controls from the study as commented by another reviewer. Instead of removing, we have elaborated the definition of healthy controls in the revised version of manuscript.

  1. Table: needs to be modifiedThere are 4 tablesthat too have not been given the right headings. The same information has been repeated in the graphs. In fact, there is no need of coloured graphs if the information is given in tables. Such colourful graphs are for power point presentations.

Response: Dear reviewer, we would like to appreciate again that you have provided us with your valuable comment to improve the manuscript. We have removed the graphs and elaborated the information in tables.

  1. Discussion:is short without any referral to other reports.

Response: Line 339-346, 381-410: The discussion section has been revised and more reports have been discussed.

  1. Conclusion: in this section, the authors state that clinical indicators for HIV-associated secondary infections were analyzed in 57 individuals (37 HIV-positive & 20 controls). Do they expect the infections in normal controls?

Response: Line 412-419: Dear reviewer, the conclusion section has been revised and rewritten.

  1. References: There are many irrelevant references and there is no uniformity. Give at least 35-40. Give the names of authors, volume and page numbers, of cited references.

Response: The reference section has been checked again and revised. New information with the relevant references has been added in the revised manuscript. More references have been cited.

  1. Acknowledgements:As this is a patient centred research article, there would be a team of Lab. Technicians and other staff doing the several investigations and other follow up tests. Please include their names in the acknowledgement section.

Response: The acknowledgment section has been revised. The people behind the research has been acknowledged in the revised version of manuscript.

  1. I would suggest the authors to read the Instructions to Authorscarefully before submitting any article to a journal. Keeping in view the impact factor of the journal, wide readership, I feel the manuscript needs major revisions at this stage. If the authors can incorporate the suggested changes in the article, it would be better. These suggestions are meant to improve the manuscript in a good journal.

Response: Dear reviewer, we really appreciate that your comments has significantly improved the quality of current manuscript. We would also like to apologise that in the previous version of manuscript, the instructions for authors have not been carefully checked and the manuscript. In the revised version of manuscript, the instructions for authors have been carefully checked.

Round 2

Reviewer 2 Report

I am impressed that the authors have incorporated all the suggestions in the manuscript quickly.

Good luck.